# Gynaecological Artificial Intelligence Diagnostics (GAID) GAID and Its Performance as a Tool for the Specialist Doctor

**DOI:** 10.3390/healthcare12020223

**Published:** 2024-01-16

**Authors:** Panayiotis Tanos, Ioannis Yiangou, Giorgos Prokopiou, Antonis Kakas, Vasilios Tanos

**Affiliations:** 1Institute of Applied Health Sciences, University of Aberdeen, NHS Grampian, Aberdeen AB24 3FX, UK; 2Department of Computer Science, University of Cyprus, Nicosia 1678, Cyprus; 3Medical School, Nicosia of University, Nicosia 2408, Cyprus; 4Aretaeio Hospital, 55-57 Andreas Avraamides, Strovolos, Nicosia 2024, Cyprus

**Keywords:** artificial intelligence, gynaecology, diagnosis, management, explainable diagnostics

## Abstract

Background: Human-centric artificial intelligence (HCAI) aims to provide support systems that can act as peer companions to an expert in a specific domain, by simulating their way of thinking and decision-making in solving real-life problems. The gynaecological artificial intelligence diagnostics (GAID) assistant is such a system. Based on artificial intelligence (AI) argumentation technology, it was developed to incorporate, as much as possible, a complete representation of the medical knowledge in gynaecology and to become a real-life tool that will practically enhance the quality of healthcare services and reduce stress for the clinician. Our study aimed to evaluate GAIDS’ efficacy and accuracy in assisting the working expert gynaecologist during day-to-day clinical practice. Methods: Knowledge-based systems utilize a knowledge base (theory) which holds evidence-based rules (“IF-THEN” statements) that are used to prove whether a conclusion (such as a disease, medication or treatment) is possible or not, given a set of input data. This approach uses argumentation frameworks, where rules act as claims that support a specific decision (arguments) and argue for its dominance over others. The result is a set of admissible arguments which support the final decision and explain its cause. Results: Based on seven different subcategories of gynaecological presentations—bleeding, endocrinology, cancer, pelvic pain, urogynaecology, sexually transmitted infections and vulva pathology in fifty patients—GAID demonstrates an average overall closeness accuracy of zero point eighty-seven. Since the system provides explanations for supporting a diagnosis against other possible diseases, this evaluation process further allowed for a learning process of modular improvement in the system of the diagnostic discrepancies between the system and the specialist. Conclusions: GAID successfully demonstrates an average accuracy of zero point eighty-seven when measuring the closeness of the system’s diagnosis to that of the senior consultant. The system further provides meaningful and helpful explanations for its diagnoses that can help clinicians to develop an increasing level of trust towards the system. It also provides a practical database, which can be used as a structured history-taking assistant and a friendly, patient record-keeper, while improving precision by providing a full list of differential diagnoses. Importantly, the design and implementation of the system facilitates its continuous development with a set methodology that allows minimal revision of the system in the face of new information. Further large-scale studies are required to evaluate GAID more thoroughly and to identify its limiting boundaries.

## 1. Introduction

Healthcare consists of medical and surgical decisions grounded upon science-based evidence [1]. Medical research is thriving and so is the knowledge we acquire from it. Inevitably, we are reaching the stage where knowledge is growing significantly, at levels that are beyond the human capability of memorising and applying all together [2]. For this reason, we are moulding specialised training pathways and subspecialised experts [3]. Obstetrics and gynaecology is an area that cares for patients with multiple other primary conditions or comorbidities, often outside the doctor’s field of knowledge and expertise. At the same time, it is a specialty that cares for patients of all ages, including paediatric and geriatric patients [4,5]. Inevitably, it includes a vast field of knowledge, which progresses at a speed much faster than the specialist doctor can easily follow. This can present more challenges for non-specialised healthcare professionals and is frustrating for those in training. Simultaneously, patient numbers increase, and appointment times decrease, while demands and expectations increase and resources are becoming more expensive or running out [6].

Since the evolution of technology, we have started seeing its applications in the day-to-day life of the physician and their decision-making, as well as the surgeon’s practical approach to management. More specifically, artificial intelligence (AI) is being incorporated throughout the various phases of the healthcare journey: diagnosis, investigation and treatment [7,8]. It can have a role in assisting the healthcare professional in decision-making, in prognosis prediction and in providing safety netting for both the patient and the clinician [9].

In obstetrics and gynaecology, the uses of AI are numerous and can be as specialised and unique as the specialty branch in which AI is applied. So far, AI has been used as a tool to interpret cardiotocography and foetal heart rate, to aid in the detection of pregnancy complications, such as gestational diabetes and preterm labour, and to review discrepancies in its interpretation, with the aim of reducing maternal and infant morbidity and mortality [10,11,12,13,14,15]. Furthermore, in the field of gynaecological surgery, the use of augmented reality helps surgeons detect vital structures, thus decreasing complications, reducing operative time, and helping surgeons in training to practice in a realistic setting [16].

Human-centric artificial intelligence (HCAI) is a modern perspective on AI that guides us to build systems that resemble the expert in a specific domain, by simulating their way of thinking and decision-making to solve real-life problems. In the medical diagnostic field, HCAI systems aim to help clinicians feel more confident in their decision-making by providing a bigger picture of differential diagnoses, within or outside their specialised field, while at the same time ensuring that the over-diagnosis of common diseases is avoided and that emergency cases are not missed out.

The gynaecological artificial intelligence diagnostics (GAID) assistant is such a system. It was developed to incorporate, as much as possible, a complete representation of the medical knowledge in gynaecology and to become a real-life tool that will practically assist the trainee and specialist doctor. It consists of a systematic patient data storage network and a user-friendly interface for record-keeping during patient visits, and it assists the medical practitioner in decision-making during the history-taking and examination of the patient. GAID provides a comprehensive list of the possible differential diagnoses under the total available current and past information about the patient. Each such possible diagnosis is justified through a comprehensive explanation for its support under the available evidence. The information data used by GAID in the decision-making process are epidemiological and patient-specific, and, while current symptomatology is the main concern, the past medical and surgical history of the patient, as well as current and past pharmacological treatments, are taken into consideration. Each of these details help both GAID and the specialist doctor throughout the decision-making journey in a stepwise and structured manner. As such, the GAID system uniquely facilitates the diagnostic process, by incorporating a functionality that guides its medical practitioner user, as the patient visit progresses, to actively seek further relevant patient information. The objective of this study was to evaluate GAID and its performance as a tool for the specialist doctor.

## 2. Materials and Methods

### 2.1. AI Technology

The development of the GAID system was strongly guided by two central principles [17]. These are (a) human in the loop and (b) sustainable knowledge acquisition. The first principle requires that the system does not aim to replace or outdo the human expert, but rather to enhance the capabilities of the human. In practice, this means that GAID is designed to provide a spectrum of most possible diagnoses, rather than a single best diagnosis, each of which comes with a comprehensive explanation (Figure 1 and Figure 2). This allows the human expert to analyse for themselves the differential diagnostic possibilities, in accordance with the clinical picture and immediate needs of the patient [18]. Furthermore, following this principle of human in the loop, GAID provides guidance on further relevant information to be collected by the medical practitioner throughout a patient visit, that will help focus closely on the possible diagnosis of the patient (Figure 3). The second HCAI principle of sustainable knowledge acquisition requires that the development of an AI system is designed as a continuous process that can easily acquire relevant knowledge, either directly from the experts or through an automatic learning process. The development of the GAID system achieves this by basing its diagnostic process on the logical reasoning of argumentation and the AI argumentation technology that supports such reasoning [16,17,18].

#### 2.1.1. Knowledge Acquisition Methodology and Algorithmic Reasoning

The human cognitive nature of argumentation allows the development of a knowledge acquisition methodology where the interaction with the human expert is undertaken exclusively in the familiar language and concepts of the application, with no exposure to the technology required. This methodology, called software development through argumentation (SoDA) [19,20], represents knowledge in terms of scenario-based preferences (SBPs). These indicate a preferred subset of decisions, in the case of GAID of diagnoses, under different application scenarios. The methodology structures these SBPs in hierarchies of increasing specificity, as more information is added to the scenarios. To apply this methodology for GAID, we were guided by the process of medical clinical practice of collecting information about a patient to give us the hierarchical structure of the SBPs. Presenting the complaints of patients will give the initial simple scenarios, which will be refined incrementally through the next phases of information gathering, such as the additional current symptoms, relevant patient records and, finally, clinical examination results. The hierarchical structures of SBPs are then represented by diagnostic tables for groups of diseases, where each consequent row of the table represents increasingly more detailed scenarios of the patient information (Table 1 and Table 2). To populate these tables, we started by drawing diagnostic information about each disease from various sources, such as PubMed, British Medical Journal (BMJ), The National Institute for Health and Care Excellence Guidelines (NICE), Center for Disease Control and Prevention (CDC) and The International Federation of Gynecology and Obstetrics (FIGO), in order to form initial drafts of such diagnostic tables, with the help of a junior doctor. These tables were then evaluated and edited by a senior doctor, with comments about corrections or further information, when necessary, until, finally, the senior doctor could certify the table as correct and complete. It is important to note that the application users are not required to know any of the technical details of the argumentation technology that underlies these tables. A very short training on how these tables are related to the application of diagnostic problems suffices, as these tables are built completely in a familiar medical language.

Currently, the GAID knowledge base consists of over 4500 tables with 1400 rows. The number of different scenario conditions, with information about symptoms, patient record and clinical examination results, across all the tables is over 1000. This number of scenario parameters shows the high complexity of the diagnostic problem; namely, to decide, from any subset of the 1000 parameters, the plausible diseases from a total set of 137 diseases (see Appendix A). The GAID diagnostic tables capture the senior expert knowledge that allows us to navigate in this complex space to diagnostic solutions. GAID reasons with these tables to arrive at plausible diagnoses, using a general AI algorithm for carrying out an introspective argumentative dialectic debate between alternative possible decision choices. When the current scenario information matches the scenario information in a row of some table, the AI reasoner forms arguments for the corresponding diseases that are shown as selected in that row. These arguments are stronger than any arguments formed for the diseases not selected in that row of the table. Furthermore, arguments that are formed from other rows of the table above the selected row are weaker. In other words, arguments from lower rows are stronger than arguments from rows above them.

Arguments for different diseases are considered counterarguments of each other. Once the arguments are formed, the dialectic reasoning aims to find those diseases that are supported by arguments that are stronger. For example, in Table 1, when the current patient information contains “vaginal discharge”, we have strong arguments for all eight diseases in this group of sexually transmitted diseases, except for the diseases SP and HSV. If and when the system receives new information that describes the vaginal discharge as profuse, with a thin texture and green, then the arguments for VC and CM become weaker than those for the diseases BV, TM and NG. Similarly, if the system also learns that the discharge is frothy, then only the diseases BV and TM are supported by arguments that are stronger than the arguments for all other diseases, and, hence, these would be the plausible diagnoses. The representation of knowledge, in terms of these tables, and the utilization of this knowledge, in terms of the high-level cognitive reasoning process of argumentation, allow GAID to provide natural explanations for the plausibility of the diseases that it diagnoses [21,22]. All that is needed is to unravel the argument(s) supporting a plausible disease, together with their relative strength over other arguments (Figure 1 and Figure 2).

#### 2.1.2. Dependence on Existing Medical Knowledge

The approach also facilitates the adaptation of the system’s knowledge with additional knowledge. This can be new knowledge that has emerged from the progress of medical science, in which case it needs to be encapsulated by new tables suitably integrated with the existing ones, or it can be new information that is acquired during the deployment or evaluation of the system, where the experts complete pieces of knowledge that were missed at the earlier initial stages of building the system. In this latter case, it is easy to recognize which tables and rows are affected, and the knowledge of the system can be modularly updated by locally adapting these tables, without the need to globally reconsider the whole system.

### 2.2. Evaluation Methods

There are three main methods of evaluating an AI decision support system. The first method is the general evaluation of the system by reviewing relevant literature. This is carried out by gathering knowledge and insight on how a system should react and perform, based on given parameters and metrics. The second method is the specific evaluation of the system using expert focus groups. This means that the decisions and performance of the system are compared to those of a doctor, in order to address any issues that concern the correctness, validity and meaningfulness of the system results, as well as the level of discrepancy between two decisions for the same case (the doctor versus the system) [23,24]. In this method, the doctor is the judge of the results and is responsible for reporting the level at which the system is correct, the level at which it helped them execute their tasks and whether their decision changed because of discrepancies between their initial decision and that of the system [23,24]. The third method is the use of real patients, in order to assess the performance of the system. Their resulting pathologies and patient information can be used to evaluate the accuracy and precision of the system [25], while their critical opinion can be used to rate how satisfactory the explanations given by the diagnostic decision support system are, compared to those of the human doctor, in order to assess their levels of trust and transparency [26]. This paper extends the examination of an AI decision support system by elaborating on the initial two stages of evaluation. Additionally, it introduces and implements this evaluation following a structured approach, detailed as follows (Figure 4):A random patient file is selected from a pool of patient records. This file is used to create an annotated file, highlighting history and initial suspicions before clinical examination.A demo file is prepared, to be used for testing.The test case is executed on GAID.
GAID collects clinical symptoms;GAID collects more detailed clinical symptoms;GAID collects the results from clinical examinations and laboratory investigations;All these details feed into the knowledge acquisition.Metrics are computed and recorded.The doctor checks the testing results and gives feedback on missing and existing diseases given by GAID. This is further validated using resources such as BMJ, PubMed, NICE Guidelines, CDC and FIGO, to ensure there is no discrepancy in knowledge between the expert and the online literature.

### 2.3. Performance Parameters and Calculations

There are three phases which involve the doctor’s empirical diagnosis, including first suspicions, initial diagnosis, and final diagnosis. The system diagnosis during each phase is extracted and compared with that of the doctor at the current stage. Therefore, three metrics are collected to calculate accuracy and precision. The performance of GAID is assessed by comparing its decision-making ability on differential diagnoses, after two rounds of questioning and one round of clinical examination findings, with the decision-making ability on differential diagnoses of the specialist gynaecologist. Accuracy is used to score the system’s diagnosis, based on the doctors. This is achieved by checking how many of the predicted disorders are considered suspicious by the expert. Additionally, precision is used to calculate the consistency of the system. For example, it can be used to calculate how often the system returns the same output based on the same inputs. The score metric equally aims to see how close it scores in comparison with a specialist. The aim here is that the system’s comparison accuracy approaches the comparison accuracy between two specialists, who, presumably, do not always agree.

To measure the relative accuracy, with respect to the predictions of the senior doctor, the number of disorders that are commonly suggested by the doctor and the system are divided by the number of diseases suggested by the doctor. However, to also penalize the system, depending on the number of diseases it suggests, we create a more complex algorithm of evaluation. We calculate, first, the difference between the disorders suggested by the system and the disorders suggested by the doctor. We subtract this amount from the total number of disorders (137) incorporated in the system and we divide it by the same number. The two formulas combined create the main accuracy metric of the system, as portrayed in Equation (1). Consequently, in the instance where the system suggests that every disorder is possible while the doctor suggests only one is, the accuracy should be near zero.
(1)$$\mathrm{Mean}\;\mathrm{Accuracy}\;\mathrm{Matrix}=(\frac{C}{e})*(\frac{D-\left(s-e\right)}{D})$$

Equation (1): Main accuracy metric formula for GAID. This representation combines an equation illustrating the mean accuracy matrix used in the model, where s = number of disorders predicted possible by the system, e = number of disorders predicted possible by the expert, c = number of common disorders between the system and the expert and D = number of disorders supported by the system = 137.

### 2.4. Patient Cohort

The performance of the system was tested and its decision-making ability validated on 50 random patients. Patient selection was retrospective and random, from a database of ten thousand records of patients presenting with new symptomatic complains at the specialist obstetrician and gynaecologist. The exclusion criteria included patients presenting for review or check up and patients with known pathologies. Patient cases were grouped under the following seven different subcategories of gynaecological presentations: bleeding (colonic polyps, endometrial polyps, miscarriage, placenta abruption, placenta previa or cervical erosion), endocrinology (Cushing’s, adrenal tumour, delayed puberty or MRKS), cancer (cervical, endometrial, ovarian, vaginal or vulva), pelvic pain (appendicitis, ectopic pregnancy, adhesions, diverticulitis, ovarian cyst rupture, ovarian, torsion, ruptured corpus luteum, adenomyosis, mesenteric artery occlusion, endometriosis, adhesions, ruptured ovarian follicles, fibroids, fallopian tube torsion, leiomyoma, adnexal tumour, bowel infection or bowel obstruction), urogynaecology (nephrolithiasis or pyelonephritis), sexually transmitted infections (anogenital warts, bacterial vaginosis, hepatitis, chlamydia, trichomoniasis, vulva candidiasis, Neisseria gonorrhoeae, syphilis, herpes simplex virus or human immunodeficiency virus) and vulva pathology (vulva intraepithelial neoplasia or vulva cancer). Each disease found in each subcategory was given a unique case ID to reduce bias.

### 2.5. Ethical Guidelines

This innovative research is an ethically sound and reliable development in accordance with the Helsinki declaration and the EU Member State guidelines report with regards to the development, adoption and use of artificial intelligence (AI) technologies and applications in the healthcare sector, as well as the standards of the Cypriot National Committee on Ethical and Reliable Artificial Intelligence. No recognisable patient information was used at any stage.

## 3. Results

The system’s accuracy is illustrated with respect to the average accuracy between the three different diagnoses given by the doctor and the system, for each of the fifty diseases individually (Figure 5). Thirty two percent of cases had an accuracy metric greater or equal to 0.800 in all three stages, and 90% of cases had an accuracy metric greater than 0.500. Fourteen percent of cases had an accuracy metric greater than 0.900.

While the results during each phase showed variation, the accuracy metric score improved after each stage of the diagnostic procedure in this evaluation. Consequently, as the diagnostic journey proceeded and more information was included, the accuracy metric increased and, therefore, the accuracy score of the final diagnosis was consistently higher than the previous two scores (Figure 6). The average accuracy metric of GAID progressively increased from 0.563 to 0.685 and 0.873 in first suspicions, initial diagnosis, and final diagnosis, respectively based on all seven different subcategories of gynaecological presentations (bleeding, endocrinology, cancer, pelvic pain, urogynaecology and prolapse, sexually transmitted infections and vulva pathology in fifty patients).

Endocrinology and gynaecological cancer groups had the highest accuracy metric scores during the stage of first suspicions, at 0.718 and 0.669, respectively. Gynaecological cancer and urogynaecology and prolapse groups had the highest accuracy metric scores during initial diagnosis, with 0.680 and 0.730, respectively. Finally, urogynaecology and prolapse and vulva pathologies had the highest accuracy metric scores during final diagnosis, with 0.975 and 0.942, respectively. All seven groups showed an increase in accuracy metric scores after each stage of the diagnosis, with pelvic pain having the smallest increase in accuracy metric scores during diagnosis (Figure 7).

Looking at outliers throughout our results section, two cases were identified where the GAID diagnosis differed significantly from that of the doctor. The doctor was able to give the reason for this difference and this was easily incorporated in the knowledge of the system. Due to the highly modular structure of the knowledge, the revision was carried out “surgically” at the appropriate part of the knowledge, without the need to affect other parts. This revision increased the final accuracy of the system from 0.853 to 0.873.

## 4. Discussion

### 4.1. Artificial Intelligence in Gynaecology

AI can have a role in assisting all healthcare professionals in decision-making and can also provide safety netting. It can reduce pressure on the general practitioner, reduce the stress and anxiety of the patients needing immediate medical assistance and assist medical clinicians in the journey of differential diagnoses. At the same time, it can make sure that that over-diagnosis of common diseases is avoided and that emergency cases are pointed out [27].

### 4.2. Advantages and Disadvantages

Although AI technologies are attracting substantial attention in medical research and clinical practice, real-life implementations are still facing obstacles [27]. AI faces multiple challenges, mainly due to its application being relatively new in nature and its uses being wide-ranging. At the same time, it is a hard concept to grasp, and it includes knowledge from multiple disciplines. Assessing the efficacy of these human-centric artificial intelligence systems (HCAISs) can also be challenging. Currently, regulations lack standards in the assessment of the safety and efficacy of HCAISs. Furthermore, in order to work well, AI systems need to be continuously trained by data from clinical studies. Simultaneously, despite the need of more robust data, thus far, AI systems have proved that they can accurately provide information on a large array of patients in a clinical setting. GAID is a great example of the way HCAISs can assist specialist doctors in the process of differential diagnoses and investigations, as well as in record-keeping, organisation and safety-netting.

### 4.3. GAID

It is inarguable that technology can improve the prognosis and management of patients and reduce healthcare costs, medical errors, and diagnostic omissions. At the same time, it helps practitioners by reducing their workload and increasing their efficiency in daily practice. AI, and more specifically GAID, can successfully demonstrate their functional applications [1,2,3,4,5,6,7,8,9,10]. GAID can guide practitioners in decision-making, assist in reaching a diagnosis, and improve case management by providing a safe platform for decision-making and data storage.

GAID’s diagnostic accuracy metric was higher at final diagnosis when more information was added, which is exactly the case with the working clinician gathering data during history-taking. However, while both GAID and the expert narrow down their differential diagnosis, GAID continues to consider all alternative diagnoses throughout the decision-making process. By achieving a wider spectrum of differential diagnoses, GAID allows for the inclusion of rarer diseases. This is reflected in Figure 6. This can be particularly helpful for doctors in training and non-specialist doctors. Clinical examination results are considered strong evidence. When they are also included in the system, the diagnostic process will speed up. True false positive and false negative disorders can therefore be excluded.

For the diseases scoring less in the final round of diagnosis than the previous two, feedback was given to the experts dealing with the software, in order to identify whether the explanation was computational or knowledge based. We should bear in mind that GAID works similarly to a human brain; the more knowledge and training it receives, the better its performance will be [28,29].

### 4.4. Knowledge Revision and Refinement

The design and implementation of GAID facilitates its continuous development. A methodology for this allows for minimal and surgical revisions of the system in the face of new information. During the testing procedure, it is important not only to score the system, based on how close its diagnosis is to that of the doctor, but to also identify any missing information that will help enrich the system’s knowledge base and improve its decision-making capabilities. To achieve this goal, the test case files, which were initially filled with the doctor’s empirical diagnosis, now host the system’s diagnosis as well. To evaluate the diagnostic capabilities of the system further, one-to-one communication with the doctor would be necessary. The specialist doctor can identify missing information regarding any contraindications, which can promptly be integrated into the system [28,29]

### 4.5. Data Privacy and Security

GAID has a three-level encrypted login protocol. Additionally, it can implement a comprehensive approach to ensure the privacy and security of sensitive medical data. This encompasses encryption, compliance with healthcare regulations, access controls, audit trails, data minimization and ongoing security measures. This commitment is designed to instil confidence in users and stakeholders regarding the protection of sensitive healthcare information. Specifically, it complies with the Health Insurance Portability and Accountability Act (HIPA) and can be integrated into the already existing hospital and healthcare trust encryption protocols, to safeguard the use of sensitive personal information.

### 4.6. Limitations

This retrospective cohort study has its limitations. It is a pilot study, with only fifty patients. All patients were seen by the same doctor and GAID was assessed by the same software operator. This may allow for consistency and systematic assessment but introduces variation and operator bias. The limited number of disease variations and presentations in the similar group of diseases is another limitation. However, grouping according to disease presentation probably reduced any variation bias. Despite the inclusion of almost all common gynaecological case presentations, not every presentation was included. GAID was focusing mainly on case history, symptomatology, and clinical examination. Once laboratory test results and imaging are incorporated into the system, an increase in final diagnostic accuracy is expected.

### 4.7. The Future

The assessment of the potential impact of GAID encompasses a multifaceted exploration of its effects on healthcare outcomes, patient satisfaction and clinician workload. As GAID systems become integrated into medical processes, there is optimism about the prospect of improved healthcare outcomes through enhanced diagnostics, personalized treatment plans and streamlined decision-making. Patient satisfaction stands to benefit from the efficiency and accuracy that AI brings to healthcare, leading to quicker diagnoses and tailored interventions. A comprehensive evaluation of these dynamics is vital to harness the full potential of GAID in healthcare, while preserving the core values of effective patient care and clinician well-being. 

## 5. Conclusions

GAID successfully demonstrates an average accuracy of 0.85 when measuring the closeness of the system’s diagnosis to that of a senior consultant. The system further provides meaningful and helpful explanations for its diagnoses that can help clinicians to develop an increasing level of trust towards the system. It also provides a practical database, which can be used as a structured history-taking assistant and as a friendly, patient record-keeper, while improving precision by broadening the list of differential diagnoses. Further large-scale studies are required to evaluate GAID more thoroughly and to identify its limiting boundaries. More importantly, the use of GAID in a prospective study with real-time patient management should be compared to the traditional standard care by an experienced gynaecologist.

## Figures and Tables

**Figure 1 healthcare-12-00223-f001:**
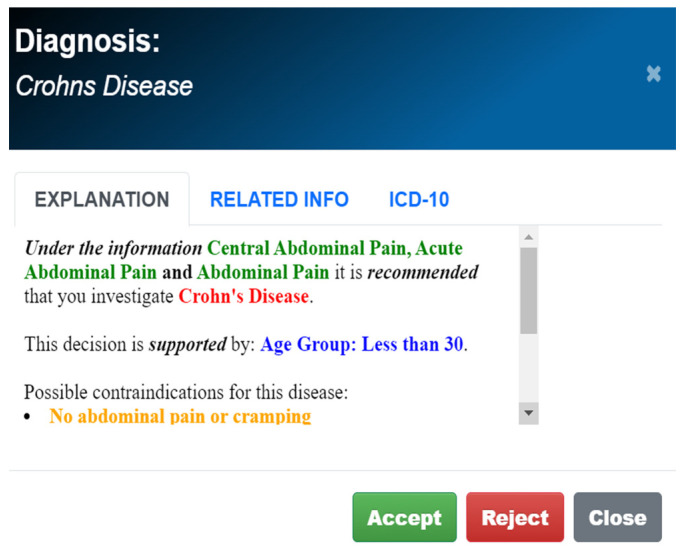
Example of a comprehensive explanation provided by GAID.

**Figure 2 healthcare-12-00223-f002:**
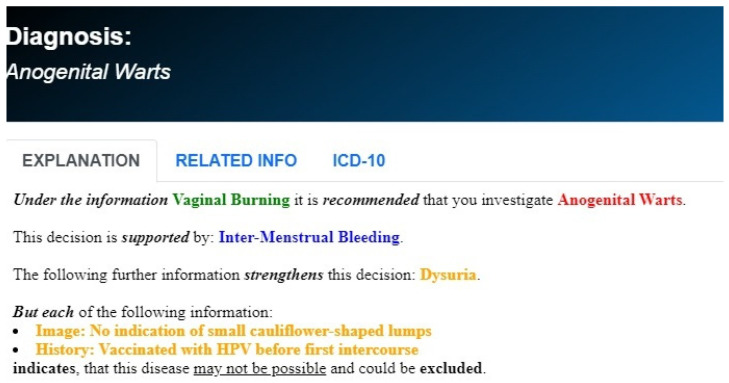
Example of s comprehensive explanation which additionally separates information that can strengthen the possibility of the disease provided by GAID.

**Figure 3 healthcare-12-00223-f003:**
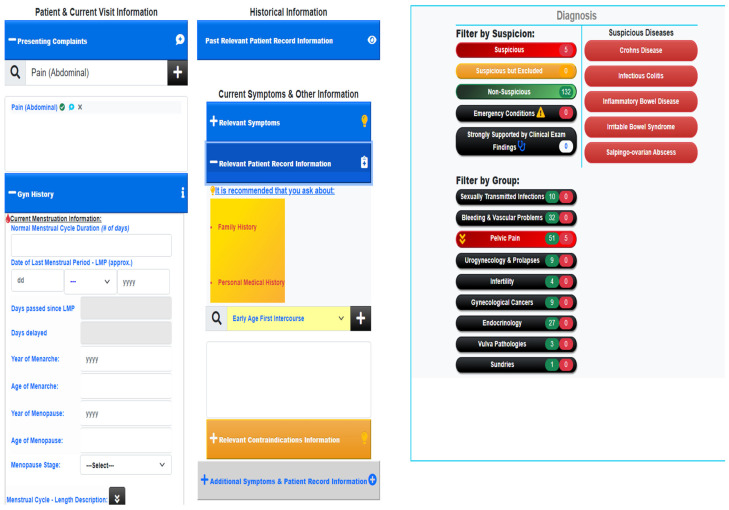
Guidance on further relevant information to be collected by the medical practitioner. Simple interface, consisting of one single page and drop-down choices to be filled by user. Diagnosis list indicating suspicions provided is changing while more information is added.

**Figure 4 healthcare-12-00223-f004:**
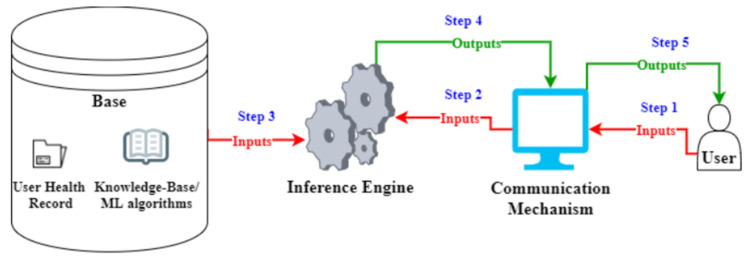
Sample architecture of a clinical decision support system (CDSS). CDSS is designed with an architecture that is based on three main components: (1) base: all the available data, plus the rules residing in the knowledge base. (2) Inference engine: runs the algorithms from the base, using the patient’s data, and outputs the results. (3) Communication mechanism: the user interface where inputs are given and outputs are presented. ML = Machine Learning.

**Figure 5 healthcare-12-00223-f005:**
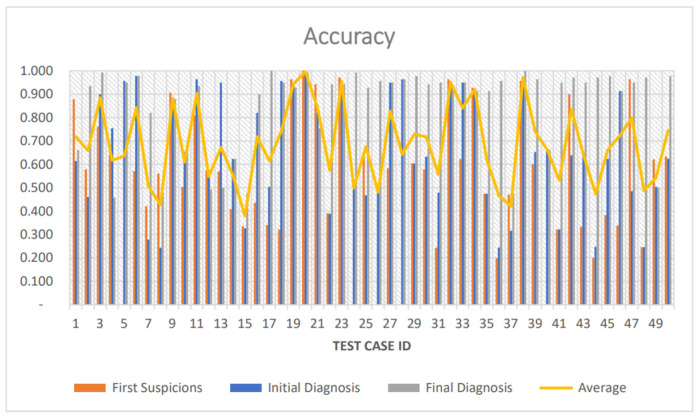
Average accuracy metric of 50 cases after first suspicions, initial diagnosis, and final diagnosis.

**Figure 6 healthcare-12-00223-f006:**
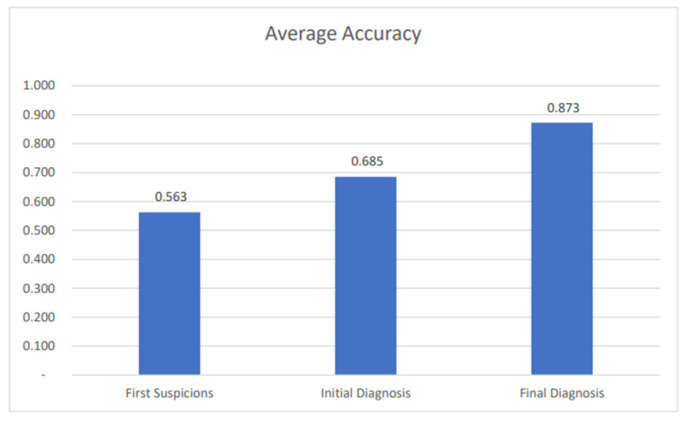
Progressive increase in accuracy metric of GAID in 50 cases; after first suspicions, initial diagnosis, and final diagnosis.

**Figure 7 healthcare-12-00223-f007:**
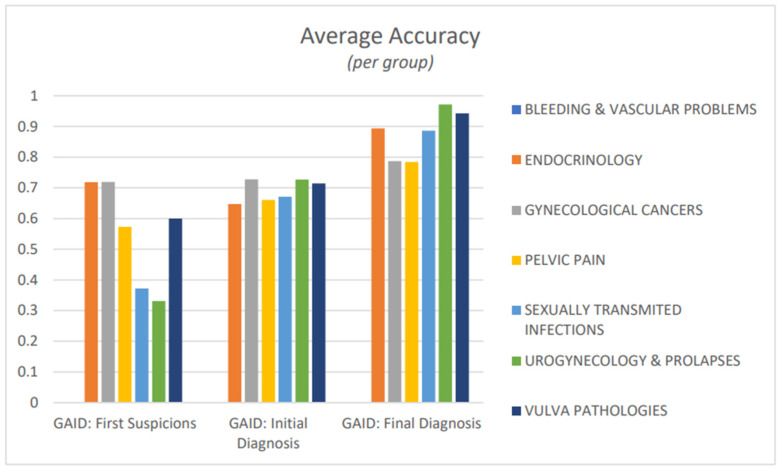
Accuracy metric of disease groups after first suspicions, initial diagnosis, and final diagnosis.

**Table 1 healthcare-12-00223-t001:** Initial/presenting symptoms: vaginal discharge.

Scenario	BV	TM	VC	SP	NG	CM	HSV	AW
Vaginal discharge (VD)	√	√	√		√	√		
Vaginal discharge (VD)	√	√			√			
++								
(quantity (VD, profuse),								
Texture (VD, thin)								
Colour (VD, green)								
Vaginal discharge (VD)	√	√						
++								
(quantity (VD, profuse),								
Texture (VD, thin)								
Colour (VD, green)								
++								
Texture (VD, frothy)								

Example of a simple diagnostic table. Each consequent row represents increasingly more detailed scenarios of patient information, leading to a more specific disease diagnosis scenario. Hierarchy: conclusion is bacterial vaginosis and/or trichomoniasis (through verbal diagnosis and observations). (BV = bacterial vaginosis, TM = trichomoniasis, VC = vulva candidiasis, SP = syphilis, NG = Neisseria gonorrhoeae, CM = chlamydia, HSV = herpes simplex virus II, AW = anogenital warts, HP = hepatitis). ++ (additional statement strongly added to previous statement, increasing likelihood of pathology with a check mark symbol √).

**Table 2 healthcare-12-00223-t002:** Initial/presenting symptoms: burning/itching.

Scenario	BV	TM	VC	SP	NG	CM	HSV	AW	HIV
Burning +/ itching	√	√	√		√	√	√	√	
Burning +/ itching			√		√	√		√	
++									
Intermenstrual_bleeding +/ postcoital_bleeding									
Burning +/ itching								√	
++									
Intermenstrual_bleeding +/ postcoital_bleeding									
++									
(lumps(small_cauliflower) +/ image (2,condyloma))									

Example of a simple diagnostic table. Each consequent row represents increasingly more detailed scenarios of patient information, leading to a more specific disease diagnosis scenario. Hierarchy: burning and/or itching, along with bleeding (inter-menstrual or post-coital) and cauliflower shaped lumps leads to the conclusion of anogenital warts. (BV = bacterial vaginosis, TM = trichomoniasis, VC = vulva candidiasis, SP = syphilis, NG = Neisseria gonorrhoeae, CM = chlamydia, HSV = herpes simplex virus II, AW = anogenital warts, HP = hepatitis). ++ (additional statement strongly added to previous statement, increasing likelihood of pathology with a check mark symbol √)

## Data Availability

The data presented in this study are available on request from the corresponding author. The data are not publicly available due to sensitive information.

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
