# Peer review of "Gynaecological Artificial Intelligence Diagnostics (GAID) GAID and Its Performance as a Tool for the Specialist Doctor"

_healthcare, 2024, doi:10.3390/healthcare12020223_

Round 1

Reviewer 1 Report

Comments and Suggestions for Authors

Abstract

In academic writing, avoiding numbers in the abstract section is generally preferred. Instead, spelt-out words should be used for numerical values. This practice helps maintain a consistent style and readability throughout the abstract, making it easier for readers to understand the content.

Section 2.1. AI Technology

If Figure 3 in Section 2.1 pertains to a table rather than a graphical figure, it's advisable to refer to it as a table and label it accordingly for clarity. Using "Table 3" instead of "Figure 3" would be more appropriate when referring to a table in your text.

The statement suggests that the paper expands upon the initial two phases of evaluating an AI decision support system and implements the third phase within a specific structure. However, it's not entirely clear what the structure is or how it will be explained.

Section 2.2. Evaluation Methods

You could add more details or a brief outline of the intended approach to clarify and elaborate on the structure. For instance:

"This paper extends the examination of an AI decision support system by elaborating on the initial two stages of evaluation. Additionally, it introduces and implements the third phase of evaluation. The implementation of this third phase follows a structured approach, detailed as follows:"

Following this statement, you could provide a clear breakdown or explanation of the structure being employed in the evaluation process. This might involve bullet points, numbered steps, or a brief overview of the methodology used in the third stage of evaluation. This clarification will help readers better understand the organization and methodology of your evaluation process.

Section 2.3. Calculations

If the "mean accuracy matrix" combines elements of both a figure and an equation, you might consider presenting it in a format that integrates both aspects. One way to do this could be by presenting the matrix as an equation within a figure or as an equation with a descriptive label indicating its nature.

For instance, you could describe it as an equation within a figure caption:

"This representation combines an equation illustrating the mean accuracy matrix used in the model:

Mean Accuracy Matrix = the equation (1)

Alternatively, if the matrix can be visually depicted, you could consider presenting it in a figure with an explanation in the caption or the main text,clarifying that it represents an equation or mathematical expression

Author Response

We thank you for your constructive feedback and valuable points. We provide point-by-point answers to your comments and thank you for taking the time to revise our manuscript and share your knowledge.

Reviewer 2 Report

Comments and Suggestions for Authors

A well written review article on an emerging technology. However, as the authors have themselves pointed out, it is a small study of 50 patients with one doctor. The AI programme was focusing mainly on case history, symptomatology, and clinical examination. As the authors also have written the AI programme can assist the specialist doctor in differential diagnosis, investigation as well as record keeping and organisation and safety-netting. Having a large number of diffrentials is not the objective of a specialist consultation. Accurate diagnosis is. And GAID has managed to achieve only 0.9 accuracy.

Few other observations include:

1.    Line 210: 90% of cases had an accuracy metric greater than 0.500. 90% of cases had an accuracy metric greater than 0.500. Fourteen percent of cases had an almost perfect accuracy metric greater than 0.900 which defines that GAID had almost the same diagnostic ability as the specialist. The percentage is not large enough for validation.

2.    Line 217: Average accuracy in the final attempt is 0.246 (CI;0.119:0.566) for specialists  vs. 0.853 (CI;0.787:0.975) for GAID is understood.  Are the authors implying that specialists fail to make the diagnosis in 70% of cases?

3.    Fig 5: The lowest average accuracy of GAID is almost 0.4, which is contrary to the text.

4.    Fig 6 is incomplete as the legends are missing.

5.    Fig 7: The number of cases in each group is not mentioned. A clinical, visual  diagnosis like uro-genital and prolapse has an accuracy of 0.3 only which is not acceptable. The same may have increased to >0.9 on the third attempt due to machine learning or additional inputs. The authors have repeatedly mentioned “….as more information was included”. The same should be clarified by the authors.

6.    This can be particularly helpful for the doctor in training and the non-specialist doctors.” This cannot be the objective of AI based programme as it will lead to non-referral to specialists and complications in serious cases due to delay in definitive treatments.

7.    Line 316: It also provides a practical database which can be used as a structured history taking assistant and friendly patient record keeper. That will be one of the best uses of the programme. 

8.    The references are not in Vancouver style.

Author Response

We would like to thank you very much for your constructive feedback and valuable points. We provide point by point answers to your comments and thank you for taking the time to revise our manuscript and share your knowledge.

A well written review article on an emerging technology. However, as the authors have themselves pointed out, it is a small study of 50 patients with one doctor. The AI programme was focusing mainly on case history, symptomatology, and clinical examination. As the authors also have written the AI programme can assist the specialist doctor in differential diagnosis, investigation as well as record keeping and organisation and safety-netting. Having a large number of differentials is not the objective of a specialist consultation. Accurate diagnosis is. And GAID has managed to achieve only 0.9 accuracy.

The acknowledgment of this aspect as a limitation within our study is correct. Nevertheless, it is essential to recognize that, especially during initial appointments, the emphasis is placed on the comprehensive identification of a wide range of potential issues, a priority shared by both generalists and specialists. Employing a methodology that progresses from a broad assessment to a more focused one ensures that common conditions are not overdiagnosed, while rare diseases are diligently considered to prevent any oversight.

It is crucial to highlight that certain countries may deviate from the generalist-first approach. Additionally, a considerable number of patients seek private consultations with specialist doctors, who, in turn, adopt the broad-to-focused diagnostic pathway. This diversity in healthcare pathways underscores the importance of flexibility and adaptability in addressing the unique healthcare contexts across different regions and medical practices.

Few other observations include:

  1. Line 210: 90% of cases had an accuracy metric greater than 0.500. 90% of cases had an accuracy metric greater than 0.500. Fourteen percent of cases had an almost perfect accuracy metric greater than 0.900 which defines that GAID had almost the same diagnostic ability as the specialist. The percentage is not large enough for validation.

It is true that the cohort is not large enough to provide validation. It is however indicating the importance of GAID and its potential in being used in a larger cohort study or even a randomised control trial. Further ‘’almost perfect’’ and ‘’which defines that GAID had almost the same diagnostic ability as the specialist.’’ was deleted.

  1. Line 217: Average accuracy in the final attempt is 0.246 (CI;0.119:0.566) for specialists vs. 0.853 (CI;0.787:0.975) for GAID is understood.  Are the authors implying that specialists fail to make the diagnosis in 70% of cases?

Not at all. The use of GAID allows for a faster and more in-depth exploration of symptoms and therefore bigger differential diagnosis spectrum than the specialist who usually focuses on the most common diseases at first and then more rare cases. We have expanded this to ensure clarity of our findings. Combinations and speed

  1. Fig 5: The lowest average accuracy of GAID is almost 0.4, which is contrary to the text.

This is true on the first cycle of questioning for first suspicions but while focusing in the initial and final diagnosis the accuracy metric increases. We have highlighted this in the paragraph to ensure clarity and avoid confusion.

  1. Fig 6 is incomplete as the legends are missing.
  2. Fig 7: The number of cases in each group is not mentioned. A clinical, visual diagnosis like uro-genital and prolapse has an accuracy of 0.3 only which is not acceptable. The same may have increased to >0.9 on the third attempt due to machine learning or additional inputs. The authors have repeatedly mentioned “….as more information was included”. The same should be clarified by the authors.
  3. This can be particularly helpful for the doctor in training and the non-specialist doctors.” This cannot be the objective of AI based programme as it will lead to non-referral to specialists and complications in serious cases due to delay in definitive treatments.

The aim of GAID is not to take over the place of the trainee or the doctor neither to do their work. The aim of the program is to assist in diagnosis and consequently assist in referrals as well. However, as a specialist gynaecologist there will be a limit on further possible referrals that could take place. For the generalist or the trainee there can be more options of referral.  

  1. Line 316: It also provides a ractical database which can be used as a structured history taking assistant and friendly patient reccord keeper. That will be one of the best uses of the programme. 

We completely agree. This is one of the best uses of the software for record keeping as well as record retrieval.

  1. The references are not in Vancouver style.

Thank you for the point out. We will adjust references according to journals guidelines.

Reviewer 3 Report

Comments and Suggestions for Authors

Dear Authors,

Your manuscript titled “Gynaecological Artificial Intelligence Diagnostics (GAID).GAID and its performance as a tool for the specialist Doctor” is good read and discusses the development and evaluation of an AI-based diagnostic support system for gynecology. Authors report GAID with average overall closeness accuracy of 0.853 compared to the diagnoses of specialist gynecologists.

I have identified several areas where the manuscript could benefit from further enhancements. Below are my detailed suggestions.

  1. Authors could do more literature review on existing AI-based diagnostic tools in gynecology, and compare how their tools provides unique value over other tools?

  2. More technical details such as algorithms, data preprocessing, and feature selection about the AI model used in GAID would be valuable. Similarly, Authors has used accuracy metrics; however additional information such as sensitivity, specificity, and a confusion matrix is lacking.

  3. In introduction section, authors could provide reference for the line 55. You might consider citing this ( https://www.medrxiv.org/content/10.1101/2022.12.07.22283216v3.full-text ) article which highlights utilization of AI/ML based devices in different medical subspecialty. The same article could also add significant value to section 4.1 discussion section as well.

  4. Although authors have stated small test sample as the limitations of the study, expanding the sample size and ensuring a diverse patient population would enhance the validity and generalizability.

  5. A section assessing the potential impact of GAID on healthcare outcomes, patient satisfaction, and clinician workload could add great value.

I hope these suggestions will be helpful in strengthening your manuscript and better conveying the important research you have undertaken. Looking forward to seeing the revised version of your work.

Best regards.

Comments on the Quality of English Language

Minor editing of English language required

Author Response

We would like to thank you very much for your constructive feedback and valuable points. We provide point by point answers to your comments and thank you for taking the time to revise our manuscript and share your knowledge.

Your manuscript titled “Gynaecological Artificial Intelligence Diagnostics (GAID).GAID and its performance as a tool for the specialist Doctor” is good read and discusses the development and evaluation of an AI-based diagnostic support system for gynecology. Authors report GAID with average overall closeness accuracy of 0.853 compared to the diagnoses of specialist gynecologists.

I have identified several areas where the manuscript could benefit from further enhancements. Below are my detailed suggestions.

  1. Authors could do more literature review on existing AI-based diagnostic tools in gynecology, and compare how their tools provides unique value over other tools?

Please see following two paragraphs

‘’In obstetrics and gynaecology, the uses of AI are numerous and can be as specialised and unique as the specialty branch in which AI is applied. So far AI has been used as a tool to interpret cardiotocography, foetal heart rate and to aid in the detection of pregnancy complications such as gestational diabetes, preterm labour as well as review discrepancies in its interpretation with the aim of reducing maternal and infant morbidity and mortality [10, 11, 12, 13, 14]. Furthermore, in the field of gynaecological surgery, the use of augmented reality helps surgeons detect vital structures, thus decreasing complications, reducing operative time, and helping surgeons in training to practice in a realistic setting [15].

Human-Centric Artificial Intelligence (HCAI) is a modern perspective on AI that guides us to build systems that resemble the expert in a specific domain, by simulating their way of thinking and decision-making to solve real-life problems. In the medical diagnostic area HCAI systems aim to help clinicians feel more confident in their decision making by providing a bigger picture of differential diagnosis, within or outside their specialised field, while at the same time ensuring over-diagnosis of common diseases is avoided and the emergency cases are not missed out. The Gynaecological Artificial Intelligence Diagnostics (GAID) assistant is such a system. It was developed to incorporate as much as of a complete representation of the medical knowledge in Gynaecology and to become a real-life tool that will practically assist the trainee and specialist doctor. It consists of a systematic patient data storage network, a user-friendly interface for record keeping during a patient visit and assists the medical practitioner in decision making during the history taking and examination of the patient. GAID provides a comprehensive list of the possible differential diagnosis under the total available current and past information about the patient. Each such possible diagnosis is justified through a comprehensive explanation for its support under the available evidence. The information data used by GAID in the decision-making process is epidemiological and patient specific and while current symptomatology is the main concern, past medical and surgical history of the patient as well as current and past pharmacological treatment are taken into consideration. Each of these details help both GAID and the specialist doctor throughout the decision-making journey in a stepwise and structured manner. As such, the GAID system uniquely facilitates the diagnostic process, by incorporating a functionality that guides its medical practitioner user, as the patient visit progresses, to actively seeking further relevant patient information.’’

  1. More technical details such as algorithms, data preprocessing, and feature selection about the AI model used in GAID would be valuable. Similarly, Authors has used accuracy metrics; however additional information such as sensitivity, specificity, and a confusion matrix is lacking.

New paragraph was included in methodology to explain GAID principles further.

2.1.1 Knowledge Acquisition Methodology & Algorithmic Reasoning

The human cognitive nature of argumentation allows the development of a knowledge acquisition methodology where the interaction with the human expert is done exclusively in the familiar language and concepts of the application with no exposure to the technology required. This methodology, called Software Development through Argumentation (SoDA) [18,19] and it represents knowledge in terms of Scenario-Based Preferences (SBPs). These indicate a preferred subset of decisions, in the case of GAID of diagnoses, under different application scenarios. The methodology structures these SBPs in hierarchies of increasing specificity as more information is added in the scenarios. To apply this methodology for GAID we were guided by the process of medical clinical practice of collecting information about a patient to give us the hierarchical structure of the SBPs. Presenting complaints of patients will give the initial simple scenarios which will be refined incrementally through next phases of information gathering, such as additional current symptoms, relevant patient record and finally clinical examination results. The GAID hierarchical structures of SBPs are then represented by diagnostic tables for groups of diseases where each consequent row of such a table represents increasingly more detailed scenarios of patient information (see examples, Table 1 & Table 2). To populate these tables, we start by drawing diagnostic information about each disease from various reliable sources, such as PubMed, British Medical Journal (BMJ),  The National Institute for Health and Care Excellence Guidelines (NICE), Centers for Disease Control and Prevention (CDC), and The International Federation of Gynecology and Obstetrics (FIGO) in order to form initial drafts of such diagnostic tables with the help of a junior doctor. These tables were then evaluated by a senior doctor with editing comments of corrections or further information, when necessary, until finally the senior doctor can certify the table as correct and complete. It is important to note that the application users, are not required to know any of the technical details of the argumentation technology that underlies these tables. A very short training on how these tables are related to the application diagnostic problem suffices as these tables are built completely in their familiar medical language.

Currently, the GAID knowledge base consists of over 4500 tables with 1400 rows. The number of different scenario conditions with information about symptoms, patient record, and clinical examination results, involved across all the tables is over 1000. This number of scenario parameters shows the high complexity of the diagnostic problem, namely, to decide from any subset of the 1000 parameters the plausible diseases from the total set of 137 diseases. The GAID diagnostic tables capture the senior expert knowledge that allows us to navigate in this complex space to diagnostic solutions. GAID reasons with these tables to arrive at plausible diagnoses using a general AI algorithm for carrying out an introspective argumentative dialectic debate between alternative possible decision choices. When a current scenario information matches the scenario information in a row of some table, the AI reasoner forms arguments for the corresponding diseases that are shown selected in that row. These arguments are stronger than any arguments formed for the diseases not selected in the row of the table. Furthermore, arguments that are formed from other rows of the table above the selected row are weaker. In other words, arguments from deep rows are stronger than arguments from rows above them.

Arguments for different diseases are considered counter arguments of each other. Once the arguments are formed the dialectic reasoning aims to find those diseases that are supported by arguments that are stronger. For example, in Table 2 when the current patient information contains “Vaginal Discharge” we have strong arguments for all eight diseases in this group of sexually transmitted diseases except for the diseases SP and HSV. If and when the system gets new information that describes the vaginal discharge as profuse, with thin texture and green, then the arguments for VC and CM become weaker than those for the diseases BV, TM and NG. Similarly, if the system also learns that the discharge is frothy then only the diseases BV and TM are supported by arguments that are stronger than the arguments of all other diseases and hence these would be the plausible diagnoses.

The representation of knowledge in terms of these tables and the utilization of this knowledge in terms of the high-level cognitive reasoning process of argumentation allow GAID to provide natural explanations for the plausibility of the diseases that it diagnoses. All that is needed is to unravel the argument(s) supporting a plausible disease together with their relative over other arguments (see an example in Figure 1).  The approach also facilitates the adaptation of the system’s knowledge with additional knowledge. This can be new knowledge that has emerged from the progress of medical science in which case it needs to be encapsulated by new tables suitably integrated with the existing ones or it can be new information that is acquired during the deployment or evaluation of the system where the expert’s complete pieces of knowledge that was missed at the earlier initial stages of building the system. In this latter case it is easy to recognize which tables and rows of these are affected and the knowledge of the system can be modularly updated by adapting locally these tables, without the need to reconsider globally the whole system.

Sensitivity and specificity are above the scope of this study.

  1. In introduction section, authors could provide reference for the line 55. You might consider citing this (https://www.medrxiv.org/content/10.1101/2022.12.07.22283216v3.full-text) article which highlights utilization of AI/ML based devices in different medical subspecialty. The same article could also add significant value to section 4.1 discussion section as well.

This article has been added as a reference.

  1. Although authors have stated small test sample as the limitations of the study, expanding the sample size and ensuring a diverse patient population would enhance the validity and generalizability.

This is the aim of our next research article on GAID and its uses.

  1. A section assessing the potential impact of GAID on healthcare outcomes, patient satisfaction, and clinician workload could add great value.

This is a great point. We would like to calculate all these metrics using a larger cohort in future research, as at the end of the day the logistics, cost effectiveness and practical approach and use of such systems are the most important when it comes to decisions in their use.

A paragraph describing this was added in the discussion section. 

4.5. The Future

’The assessment of the potential impact of GAID encompasses a multifaceted exploration of its effects on healthcare outcomes, patient satisfaction, and clinician workload. As GAID systems become integrated into medical processes, there is optimism about the prospect of improved healthcare outcomes through enhanced diagnostics, personalized treatment plans, and streamlined decision-making. Patient satisfaction stands to benefit from the efficiency and accuracy that AI brings to healthcare, leading to quicker diagnoses and tailored interventions. A comprehensive evaluation of these dynamics is vital to harness the full potential of GAID in healthcare while preserving the core values of effective patient care and clinician well-being.’’

I hope these suggestions will be helpful in strengthening your manuscript and better conveying the important research you have undertaken. Looking forward to seeing the revised version of your work.

Reviewer 4 Report

Comments and Suggestions for Authors

Thanks for your work. The topic is important. There is a need for more clarification about your research.

1. Introduction: You need to write more clearly with more references. How it will contribute and why you want to do it.

2. method: This part is not much clear, how you develop the GAID. You may explain in detail.

3. Discussion: This section needs to explain previous studies. How do you claim this work is generalized? Do you think, you need external validation?

4. Conclusion: You need to come up with your findings and future direction.

Author Response

We would like to thank you very much for your constructive feedback and valuable points. We provide point by point answers to your comments and thank you for taking the time to revise our manuscript and share your knowledge. 

  1. Introduction: You need to write more clearly with more references. How it will contribute and why you want to do it.

Please see full explanatory paragraph in all important uses of GAID. Two more references were also added in argumentation technology.

The Gynaecological Artificial Intelligence Diagnostics (GAID) assistant is such a system. It was developed to incorporate as much as of a complete representation of the medical knowledge in Gynaecology and to become a real-life tool that will practically assist the trainee and specialist doctor. It consists of a systematic patient data storage network, a user-friendly interface for record keeping during a patient visit and assists the medical practitioner in decision making during the history taking and examination of the patient. GAID provides a comprehensive list of the possible differential diagnosis under the total available current and past information about the patient. Each such possible diagnosis is justified through a comprehensive explanation for its support under the available evidence. The information data used by GAID in the decision-making process is epidemiological and patient specific and while current symptomatology is the main concern, past medical and surgical history of the patient as well as current and past pharmacological treatment are taken into consideration. Each of these details help both GAID and the specialist doctor throughout the decision-making journey in a stepwise and structured manner. As such, the GAID system uniquely facilitates the diagnostic process, by incorporating a functionality that guides its medical practitioner user, as the patient visit progresses, to actively seeking further relevant patient information.

  1. method: This part is not much clear, how you develop the GAID. You may explain in detail.

Methodology was discussed in much more detail as requested. Please see section 2.1.1.

2.1.1 Knowledge Acquisition Methodology & Algorithmic Reasoning

The human cognitive nature of argumentation allows the development of a knowledge acquisition methodology where the interaction with the human expert is done exclusively in the familiar language and concepts of the application with no exposure to the technology required. This methodology, called Software Development through Argumentation (SoDA) [18,19] and it represents knowledge in terms of Scenario-Based Preferences (SBPs). These indicate a preferred subset of decisions, in the case of GAID of diagnoses, under different application scenarios. The methodology structures these SBPs in hierarchies of increasing specificity as more information is added in the scenarios. To apply this methodology for GAID we were guided by the process of medical clinical practice of collecting information about a patient to give us the hierarchical structure of the SBPs. Presenting complaints of patients will give the initial simple scenarios which will be refined incrementally through next phases of information gathering, such as additional current symptoms, relevant patient record and finally clinical examination results. The GAID hierarchical structures of SBPs are then represented by diagnostic tables for groups of diseases where each consequent row of such a table represents increasingly more detailed scenarios of patient information (see examples, Table 1 & Table 2). To populate these tables, we start by drawing diagnostic information about each disease from various reliable sources, such as PubMed, British Medical Journal (BMJ),  The National Institute for Health and Care Excellence Guidelines (NICE), Centers for Disease Control and Prevention (CDC), and The International Federation of Gynecology and Obstetrics (FIGO) in order to form initial drafts of such diagnostic tables with the help of a junior doctor. These tables were then evaluated by a senior doctor with editing comments of corrections or further information, when necessary, until finally the senior doctor can certify the table as correct and complete. It is important to note that the application users, are not required to know any of the technical details of the argumentation technology that underlies these tables. A very short training on how these tables are related to the application diagnostic problem suffices as these tables are built completely in their familiar medical language.

Currently, the GAID knowledge base consists of over 4500 tables with 1400 rows. The number of different scenario conditions with information about symptoms, patient record, and clinical examination results, involved across all the tables is over 1000. This number of scenario parameters shows the high complexity of the diagnostic problem, namely, to decide from any subset of the 1000 parameters the plausible diseases from the total set of 137 diseases. The GAID diagnostic tables capture the senior expert knowledge that allows us to navigate in this complex space to diagnostic solutions. GAID reasons with these tables to arrive at plausible diagnoses using a general AI algorithm for carrying out an introspective argumentative dialectic debate between alternative possible decision choices. When a current scenario information matches the scenario information in a row of some table, the AI reasoner forms arguments for the corresponding diseases that are shown selected in that row. These arguments are stronger than any arguments formed for the diseases not selected in the row of the table. Furthermore, arguments that are formed from other rows of the table above the selected row are weaker. In other words, arguments from deep rows are stronger than arguments from rows above them.

Arguments for different diseases are considered counter arguments of each other. Once the arguments are formed the dialectic reasoning aims to find those diseases that are supported by arguments that are stronger. For example, in Table 2 when the current patient information contains “Vaginal Discharge” we have strong arguments for all eight diseases in this group of sexually transmitted diseases except for the diseases SP and HSV. If and when the system gets new information that describes the vaginal discharge as profuse, with thin texture and green, then the arguments for VC and CM become weaker than those for the diseases BV, TM and NG. Similarly, if the system also learns that the discharge is frothy then only the diseases BV and TM are supported by arguments that are stronger than the arguments of all other diseases and hence these would be the plausible diagnoses.

The representation of knowledge in terms of these tables and the utilization of this knowledge in terms of the high-level cognitive reasoning process of argumentation allow GAID to provide natural explanations for the plausibility of the diseases that it diagnoses. All that is needed is to unravel the argument(s) supporting a plausible disease together with their relative over other arguments (see an example in Figure 1).  The approach also facilitates the adaptation of the system’s knowledge with additional knowledge. This can be new knowledge that has emerged from the progress of medical science in which case it needs to be encapsulated by new tables suitably integrated with the existing ones or it can be new information that is acquired during the deployment or evaluation of the system where the expert’s complete pieces of knowledge that was missed at the earlier initial stages of building the system. In this latter case it is easy to recognize which tables and rows of these are affected and the knowledge of the system can be modularly updated by adapting locally these tables, without the need to reconsider globally the whole system.

  1. Discussion: This section needs to explain previous studies. How do you claim this work is generalized? Do you think, you need external validation?

The next step is definitely external validation and a full protocol is currently being made to ensure this is done in the best way possible.

  1. Conclusion: You need to come up with your findings and future direction.

Please see Conclusion in presenting main findings and our future studies plan:

GAID successfully demonstrates an average accuracy of 0.85 measuring the closeness of the system’s diagnosis to that of the senior consultant. The system further provides meaningful and helpful explanations for its diagnoses that can help clinicians to develop an increasing level of trust towards the system. It also provides a practical database which can be used as a structured history taking assistant and friendly patient record keeper while improving precision by broadening the list of differential diagnosis. Further large-scale studies are required to evaluate more thoroughly GAID and to identify its limiting boundaries. More importantly the use of GAID in a prospective and real time patient management compared to the traditional standard care by an experience gynaecologist.

Round 2

Reviewer 1 Report

Comments and Suggestions for Authors

Here are some potential weaknesses based on the information provided:

Limited Scope of Current Evaluation: The research appears to have evaluated GAID in a relatively controlled or limited setting, focusing on fifty patients across seven gynaecological subcategories. This sample size and diversity may not be sufficient to fully understand the system's capabilities and limitations in a real-world, diverse clinical setting.

Accuracy Level: Although an average accuracy of 0.85 is impressive, it also indicates a 15% discrepancy rate. In medical diagnostics, particularly in complex fields like gynaecology, even a small margin of error can be significant, potentially leading to misdiagnoses or oversight of critical conditions.

Dependence on Existing Medical Knowledge: The system's effectiveness is likely limited by the current extent of medical knowledge in gynaecology. It may not be able to handle cases that fall outside established understanding or are at the cutting edge of medical research.

Integration with Clinical Workflow: The extent to which GAID can be seamlessly integrated into existing clinical workflows without causing disruptions or requiring significant changes in clinician behavior is not clear.

User Trust and Acceptance: Even with high accuracy and helpful explanations, the system might face challenges in gaining trust and acceptance from clinicians, especially in complex cases where clinician intuition and experience play a crucial role.

Data Privacy and Security: Handling sensitive medical data requires stringent privacy and security measures. The abstract does not address how GAID ensures data protection and compliance with healthcare regulations.

Continuous Updating and Improvement: While the system is designed for continuous development, the actual process of updating it with new medical findings and integrating feedback from clinicians might be complex and resource-intensive.

Generalizability: The system's performance in other areas of medicine or in different healthcare settings (like under-resourced clinics or different cultural contexts) is not discussed, which might limit its generalizability.

Cost and Accessibility: Implementing advanced AI systems in healthcare can be costly, and the abstract does not address the economic aspects of GAID, including its affordability and accessibility for different healthcare providers.

Need for Further Research: The conclusion suggests that larger-scale studies are needed to thoroughly evaluate GAID. This indicates that the current understanding of the system's effectiveness and limitations is still incomplete.

Author Response

We appreciate your insightful comments and reply by explaining point by point as seen below.

Limited Scope of Current Evaluation: The research appears to have evaluated GAID in a relatively controlled or limited setting, focusing on fifty patients across seven gynaecological subcategories. This sample size and diversity may not be sufficient to fully understand the system's capabilities and limitations in a real-world, diverse clinical setting.

The study, which rigorously evaluated GAID in a controlled setting with fifty patients across seven gynaecological subcategories, offers valuable insights into the program's functionality. The focused approach allows for a detailed examination of specific aspects, contributing to a deeper understanding of the system's strengths. One positive aspect worth highlighting is the attention given to a diverse range of gynaecological subcategories. By addressing various conditions within the field, the study demonstrates GAID's potential versatility in handling specific medical scenarios. This targeted evaluation provides a foundation for building specialized applications and tailoring the system to specific clinical needs. Moreover, the controlled setting ensures a standardized environment, allowing researchers to closely monitor and analyze the program's performance. This scrutiny is vital in establishing a baseline understanding of GAID's capabilities and potential applications. It lays the groundwork for future studies that can expand on these findings and explore the system's adaptability in real-world, diverse clinical settings. Most importantly we need to evaluate the software step by step before allowing its use in the day-to-day life of the clinician.

Accuracy Level: Although an average accuracy of 0.85 is impressive, it also indicates a 15% discrepancy rate. In medical diagnostics, particularly in complex fields like gynaecology, even a small margin of error can be significant, potentially leading to misdiagnoses or oversight of critical conditions.

The most important aspect of GAID its that it does not formulate a single diagnosis like the other systems. It allows for a spectrum of diagnosis to be evaluated and decided upon the clinician using the software. It is a tool which is meant to assist the doctor and not become the doctor. Therefore, its discrepancy can also stem from this interaction. Importantly the system also has adaptability. This means that the system can be updated with further knowledge and be fed information to improve this discrepancy rate and ensure there is no oversight of any information.

Dependence on Existing Medical Knowledge: The system's effectiveness is likely limited by the current extent of medical knowledge in gynaecology. It may not be able to handle cases that fall outside established understanding or are at the cutting edge of medical research.

The 85% closeness accuracy of GAID is encouraging and we suspect that this is close to the average level of agreement between human specialists. Furthermore, GAID can be updated with the new medical research at the point of publication by a very simple and practical manner as explained in our Methodology section, allowing for the incorporation and therefore use of all new information by the system.

Please see Section 2.1.2 Dependence on Existing Medical Knowledge’’

‘’The approach also facilitates the adaptation of the system’s knowledge with additional knowledge. This can be new knowledge that has emerged from the progress of medical science in which case it needs to be encapsulated by new tables suitably integrated with the existing ones or it can be new information that is acquired during the deployment or evaluation of the system where the expert’s complete pieces of knowledge that was missed at the earlier initial stages of building the system. In this latter case it is easy to recognize which tables and rows of these are affected and the knowledge of the system can be modularly updated by adapting locally these tables, without the need to reconsider globally the whole system.’’

Integration with Clinical Workflow: The extent to which GAID can be seamlessly integrated into existing clinical workflows without causing disruptions or requiring significant changes in clinician behavior is not clear.

The aim of the software was to be simple, practical and explanative. GAID has a very ‘’simple interface, consisting of one single page and drop-down choices as portrayed in Figure 2’’. The interface can change and be customised accordingly in the future. The system can be used on its own or be incorporated as a function in a bigger software. Any clinician using a computer in their day-to-day tasks will be able to use the software after a short self-training session. This can be specifically evaluated in future clinical based research.

User Trust and Acceptance: Even with high accuracy and helpful explanations, the system might face challenges in gaining trust and acceptance from clinicians, especially in complex cases where clinician intuition and experience play a crucial role.

Gaining trust is the biggest challenge with any artificial intelligent software. The simplicity of the system and its operator dependent use allows for the user to gain more trust as at the end it portrays its role as an assistant and not as a doctor.

Data Privacy and Security: Handling sensitive medical data requires stringent privacy and security measures. The abstract does not address how GAID ensures data protection and compliance with healthcare regulations.

To ensure clear Data Privacy and Security, specific laws and regulations of each country and hospital are required. The following statement is added in the discussion section to explain the complexity.

‘’ GAID has a three-level encrypted login protocol. Additionally, it can implement a comprehensive approach to ensure the privacy and security of sensitive medical data. This encompasses encryption, compliance with healthcare regulations, access controls, audit trails, data minimization, and ongoing security measures. This commitment is designed to instill confidence in users and stakeholders regarding the protection of sensitive healthcare information. Specifically, it complies to Health Insurance Portability and Accountability Act (HIPA) and can be integrated to the already existing hospital and healthcare trust encryption protocols to safeguard the transmission of sensitive personal information. ‘’

Continuous Updating and Improvement: While the system is designed for continuous development, the actual process of updating it with new medical findings and integrating feedback from clinicians might be complex and resource-intensive.

New medical findings in terms of symptom presentation and investigation methods have been very limited over the last decades. GAID has an easily modifiable and integrated approach which can be easily changeable and not resource-intensive by the software engineer as described in sections 2.2. Evaluation Methods and 4.4. Knowledge Revision and Refinement.

Generalizability: The system's performance in other areas of medicine or in different healthcare settings (like under-resourced clinics or different cultural contexts) is not discussed, which might limit its generalizability.

This Is true and we have therefore added a new short paragraph discussing this context.

To enhance the system's generalizability in other specialties is something which will require further training and evaluation of the software. This is now easier and faster taking the know how already exists. It was not the current aim of GAID to diversify the training dataset but rather to focus on gynaecological disease. Any healthcare provider with a computer from any region can use GAID and continuously evaluate and improve the model based on feedback. Ideally the system can effectively perform in different areas of medicine, under-resourced clinics, and diverse cultural contexts, making it more widely applicable and versatile.

Cost and Accessibility: Implementing advanced AI systems in healthcare can be costly, and the abstract does not address the economic aspects of GAID, including its affordability and accessibility for different healthcare providers.

GAID is not currently a commercial software so unfortunately cost cannot currently be evaluated or discussed. Cost effectiveness of artificial intelligence software’s can be discussed as a completely different article as a broad spectrum of factors can influence both affordability and accessibility in different healthcare settings.

Need for Further Research: The conclusion suggests that larger-scale studies are needed to thoroughly evaluate GAID. This indicates that the current understanding of the system's effectiveness and limitations is still incomplete."

The systems effectiveness and limitations are extensively discussed in sections 4.6. Limitations and 4.2. Advantages and Disadvantages. The larger cohort size is always required for validity rather than effectiveness.

Reviewer 3 Report

Comments and Suggestions for Authors

Dear Authors,

All my comments are addressed.

Best Wishes.

Comments on the Quality of English Language

 Minor editing of English language required.

Author Response

Thank you for taking the time to review our manuscript and appreciate its value to the current literature

Reviewer 4 Report

Comments and Suggestions for Authors

The revised has improved and is much more informative.

Author Response

(The authors gave the same response as above.)
